# Enhancement of Polyvinyl Acetate (PVAc) Adhesion Performance by SiO$_2$ and TiO$_2$ Nanoparticles

**Gorana Petković \***, **Marina Vukoje \***, **Josip Bota** and **Suzana Pasanec Preprotić**

Faculty of Graphic Arts, University of Zagreb, Getaldićeva 2, 10000 Zagreb, Croatia; jbota@grf.hr (J.B.); spasanec@grf.hr (S.P.P.)

**\*** Correspondence: gorana.petkovic@grf.hr (G.P.); marina.vukoje@grf.hr (M.V.);
Tel.: +385-(0)-1-23-71-080/216 (G.P.); +385-(0)-1-23-71-080/120 (M.V.)

**Abstract:** Post press processes include various types of bonding and adhesives, depending upon the nature of adherends, the end use performance requirements and the adhesive bonding processes. Polyvinyl acetate (PVAc) adhesive is a widely used adhesive in the graphic industry for paper, board, leather and cloth. In this study, the enhancement of PVAc adhesion performance by adding different concentrations (1%, 2% and 3%) of silica (SiO$_2$) and titanium dioxide (TiO$_2$) nanoparticles was investigated. The morphology of investigated paper-adhesive samples was analyzed by SEM microscopy and FTIR spectroscopy. In addition, the optimal adhesion at the interface of paper and adhesive was found according to calculated adhesion parameters by contact angle measurements (work of adhesion, surface free energy of interphase, wetting coefficient). According to obtained surface free energy (SFE) results, optimum nanoparticles concentration was 1%. The wettability of the paper-adhesive surface and low SFE of interphase turned out as a key for a good adhesion performance. The end use T-peel resistance test of adhesive joints confirmed enhancement of adhesion performance. The highest strength improvement was achieved with 1% of SiO$_2$ nanoparticles in PVAc adhesive.

**Keywords:** PVAc adhesive; nano-enhanced adhesive; adhesive joint; adhesion performance; peel resistance

## 1. Introduction

Adhesives are non-metal materials that are used to join two or more components together through attractive forces acting across the interface. One of the main features of all adhesives is the relatively small amount needed to form a joint between two substrates compared to the weight of the final product. Selection of adhesive type and form depends upon the nature of the adherents, the end use performance requirements and the adhesive bonding processes [1–3]. Graphic production, especially post press processes, include various types of bonding and adhesives. There is no unique adhesive that can fulfil all post press graphic applications and it is usually necessary to compromise when selecting a practical adhesive system [4]. Polyvinyl acetate adhesives (PVAc) are customized for the short runs of graphic production or production of personalized products on demand [1,5]. PVAc is water-based adhesive and it is considered more environmentally acceptable compared to solvent-based adhesives. Therefore, current guidelines for development of adhesive technology are focused on replacing the solvent-based adhesives with water-based, as well as achieving comparable or better quality of adhesive joints for particular application [6].

Polyvinyl acetate is a clear, water-white, thermoplastic synthetic resin produced from its monomer by emulsion polymerization. PVAc is good adhesive for paper, plastics, metal foil, leather, cloth and wood, but it is also used as a general building adhesive [1,7,8]. PVAc sets through evaporation and

diffusion of the water into the substrate, and at the same time, by polymerization of polymer particles as the water evaporates. PVAc may be applied through different methods, such as brushing, flowing, spraying, roll coating, knife coating or silk screening [1,9]. The main advantages of PVAc are easy and wide application, elasticity, resistance to aging, low cost and availability, resistance to bacterial and fungicidal attack and non-toxicity [1,10–14]. The main disadvantages of PVAc are low resistance to weather and moisture, poor resistance to most solvents, slow curing and setting speed, and creeping under substantial static load [1,9,14,15].

In order to overcome these PVAc disadvantages numerous studies on PVAc adhesives modification, by adding nanoparticles, were carried out over the last few years. The researchers introduced nano clay (NC), cellulose nanofibrils (CNFs), silica ($SiO_2$) and titanium dioxide ($TiO_2$) nanoparticles [16–25]. Due to a small size and large surface area of the nanosized particles, only small amounts are needed to make significant changes in adhesive performance. For each particular adhesive application there is always an optimal concentration of nanofillers for achieving the best adhesive performance [16,26,27]. In addition, a good dispersion of nanofillers is needed to achieve the best performing nanocomposite. According to previous researches, more effective bonding properties and thermal stability of PVAc adhesive can be achieved by adding $SiO_2$ and $TiO_2$ nanoparticles [16,18,21,22], as well as NC and CNFs nanofillers [19,20,24]. By using nanoparticles as stabilizer, the water resistant property of the adhesive film can be improved significantly, as well as their mechanical properties and adhesion performance when in increased moisture and temperature conditions. Nanoparticles distributed in the polymer matrix change the effective diffusion path length and thus increase the water vapor barrier properties [21,23]. Polymer nanocomposites are becoming very important hybrids in a variety of industries, as they combine desirable properties of nanoparticles (mechanical and thermal stability, water resistance, durability, etc.) with desirable properties of investigated polymer [25,28]. Based on previous research, it can be concluded that modification of PVAc adhesive with NC, CNFs, $SiO_2$ or $TiO_2$ nanoparticles has not been explored enough for the purpose of post press processes in graphic production. The existing studies are mainly related with the improvement of bonding performance in wood and furniture industry [16,18,21,23,24,27]. Considering that the performance of adhesive depends upon the nature of adherends [26], the existing results cannot be directly linked to the adhesive systems within the graphic production, more precisely, within the production of high quality paper products.

The aim of this study was the modification of PVAc adhesive with 1%, 2% and 3% of $SiO_2$ and $TiO_2$ nanoparticles on performance by analyzing the surface free energy (SFE) of dry film and adhesion performance of paper-adhesive samples. In addition to SFE determination, Scanning Electron Microscopy (SEM) and Fourier Transform Infrared (FTIR) Spectroscopy, a T-peel resistance test was also carried out. In this research, $SiO_2$ and $TiO_2$ nanoparticles were selected according to recommendations and results of previous studies on polymer matrix nanocomposites (PMCs) [16–19,21–25,27]. According to their remarkable properties and acceptable costs, $SiO_2$ nanoparticles are recommended for the production of high-performance adhesives and coatings. In addition, they can be used for the enhancement of mechanical strength, flexibility, and durability, as well as modification of rheological properties of liquids, adhesives and elastomers [17,25,27]. $TiO_2$ nanoparticles have been rarely used in PMCs modification studies, due to weaker mechanical strength results compared to $SiO_2$ nanoparticles. However, the addition of $TiO_2$ nanoparticles can reduce material degradation under the influence of UV radiation, ensure consistence of coloration and to increase the lifetime of the final products [25]. The invisibility or whiteness of the adhesive bond line can be very important for the appearance of the final graphic product. $TiO_2$ pigmentary properties impart whiteness, brightness and opacity when incorporated into PMCs. By increasing the lifetime of the final product, adhesive bonding processes can become even more important in graphic post press processes for the production of more competitive and more durable products compared to other joining methods (e.g., sewing, stapling, riveting). NC is still the most studied nanomaterial due to low cost production and availability, but mostly it is used in the construction industry for modification of concrete, cement, asphalt and bitumen

to enhance their barrier, mechanical and rheological properties, fire retardancy, as well as their liquid infusion resistant properties [24,25]. Addition of NC or CNFs can cause an increase in viscosity [19,25]. Moreover, addition of NC can cause yellowing [24]. The mentioned changes are not desirable when using PVAc adhesive in graphic post press processes, regardless of the application methods and appearance of the final product. It is expected that the advantages obtained by addition of relative expensive nanoparticles will compensate the increase of the adhesive price [26]. Therefore, it is of great interest to keep the share of nanoparticles in the selected adhesive as small as possible. According to previous studies, it is possible to achieve the best performance of selected PMCs with a smaller share of $SiO_2$ or $TiO_2$ nanoparticles (1%) [16] compared to NC (4%) [18] and CNFs (10%) [19]. Considering the fact that the results of conducted studies are not related to PVAc adhesive systems and mutually differ, higher increase in mechanical strength of adhesives can be achieved with $SiO_2$ nanoparticles than NC or CNFs. For example, the epoxy adhesive mechanical strength was increased up to 66% with $SiO_2$ nanoparticles, with CNFs up to 5% and with NC up to 7%, respectively [26]. The mechanical strength of polyurethane adhesive was increased up to 462% with $SiO_2$ nanoparticles compared to 68% with NC, while the increase of acrylic adhesive was up to 219% with $SiO_2$ nanoparticles compared to 146% with NC [26]. By addition of $SiO_2$ and $TiO_2$ nanoparticles, besides mechanical properties, thermal stability and durability improvement, considerable increase of the bonding strength of PVAc adhesive, at open time 5 and 10 min, can be achieved as well [21], which can lead to higher efficiency of bonding processes in graphic production.

## 2. Materials and Methods

### 2.1. Materials

#### 2.1.1. Paper

For the evaluation of adhesion performance two office papers (Navigator Universal (**A**), Royal White (**B**)) were used. Both papers are made from primary fibers and have the same grammage (80 g/m$^2$). They are produced by different paper manufacturers reachable on market and have a different price range (A:B = 1.4:1). Papers were characterized according to standard methods: roughness (ISO 4287:1997) [29], moisture (T 412 om–16) [30], $CaCO_3$ content (T 553 om–15) [31], ash content (T 413 om–17) [32], absorptivity—Cobb test (T 441 om–13) [33] and tensile breaking strength (ISO 1924–2:2008) [34].

#### 2.1.2. Adhesive

In this research, adhesion performance of polyvinyl acetate (Signokol L) and nano-enhanced polyvinyl acetate adhesive was investigated. Signokol L is water dispersion of vinyl acetate homopolymers with polyvinyl alcohol and addition of plasticizers [35]. Properties of the used adhesive are listed in Table 1.

**Table 1.** Properties of Signokol L adhesive given by the producer [35].

| State of Matter: | Liquid |
|---|---|
| Main Purpose: | paper, board |
| Color: | white |
| Dry Film Color: | transparent |
| Density (20 °C): | 1.0776 g/cm$^3$ |
| Viscosity (20 °C): | 8–10 Pa s |
| pH Value: | 6 ± 0.5 |
| Solid Content: | 45 ± 2% |

### 2.1.3. Nanoparticles

Silica (SiO$_2$) (Aerosil R 8200) and titanium dioxide (TiO$_2$) (Aeroxide P25) nanoparticles were used for PVAc adhesive modifications. Both are odorless, solid, white powders with approximately the same temped density (140 g/L), but a different BET surface area (135–185 m$^2$/g SiO$_2$; 35–65 m$^2$/g TiO$_2$) and assay based on ignited material (≥99.8% SiO$_2$; ≥99.5% TiO$_2$) [36,37].

### 2.2. Modification of PVAc Adhesive

For evaluation of adhesion performance seven samples of adhesives were used—PVAc adhesive and nano-enhanced PVAc adhesives modified with 1%, 2% and 3% (based on solid mass of PVAc) of SiO$_2$ and TiO$_2$ nanoparticles. The defined amounts of nanoparticles were mixed with the PVAc adhesive using the IKA T 25 digital Ultra-disperser (IKA-Werke, Staufen, Germany) (Figure 1).

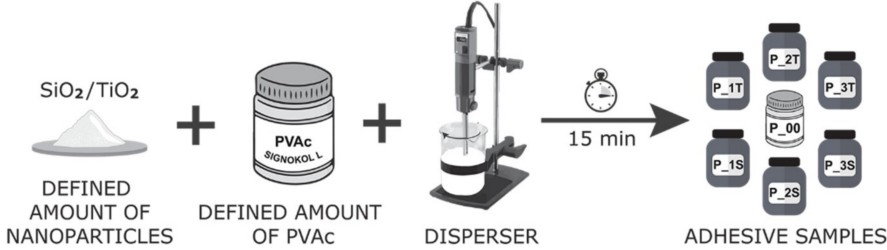

**Figure 1.** Mixing procedure for polyvinyl acetate (PVAc) modification with defined amount of SiO$_2$ and TiO$_2$ nanoparticles.

To achieve uniform dispersion the compounds were mixed 15 min. The frequency was gradually increased during the first 5 min from 0 to 7000 rpm then retaining that frequency in the next 10 min. All prepared nano-enhanced adhesives were produced under the same conditions, using the same mixing procedure. The list of used adhesives is in Table 2.

**Table 2.** List of used adhesive, paper-adhesive and adhesive joint samples.

| Sample | Abbreviation | Description |
|---|---|---|
| Adhesive | P_00 | PVAc |
| | P_1S | PVAc + 1% SiO$_2$ |
| | P_2S | PVAc + 2% SiO$_2$ |
| | P_3S | PVAc + 3% SiO$_2$ |
| | P_1T | PVAc + 1% TiO$_2$ |
| | P_2T | PVAc + 2% TiO$_2$ |
| | P_3T | PVAc + 3% TiO$_2$ |
| Paper-adhesive | P_00/A | paper A + PVAc |
| | P_1S/A | paper A + PVAc + 1% SiO$_2$ |
| | P_2S/A | paper A + PVAc + 2% SiO$_2$ |
| | P_3S/A | paper A + PVAc + 3% SiO$_2$ |
| | P_1T/A | paper A + PVAc + 1% TiO$_2$ |
| | P_2T/A | paper A + PVAc + 2% TiO$_2$ |
| | P_3T/A | paper A + PVAc + 3% TiO$_2$ |
| | P_00/B | paper B + PVAc |
| | P_1S/B | paper B + PVAc + 1% SiO$_2$ |
| | P_2S/B | paper B + PVAc + 2% SiO$_2$ |
| | P_3S/B | paper B + PVAc + 3% SiO$_2$ |
| | P_1T/B | paper B + PVAc + 1% TiO$_2$ |
| | P_2T/B | paper B + PVAc + 2% TiO$_2$ |
| | P_3T/B | paper B + PVAc + 3% TiO$_2$ |

**Table 2.** *Cont.*

| Sample | Abbreviation | Description |
|---|---|---|
| Adhesive joint | A/P_00/A | paper A + PVAc + paper A |
| | A/P_1S/A | paper A + PVAc + 1% $SiO_2$ + paper A |
| | A/P_1T/A | paper A + PVAc + 1% $TiO_2$ + paper A |
| | B/P_00/B | paper B + PVAc + paper B |
| | B/P_1S/B | paper B + PVAc + 1% $SiO_2$ + paper B |
| | B/P_1T/B | paper B + PVAc + 1% $TiO_2$ + paper B |

*2.3. Test Samples Preparation*

2.3.1. Paper-Adhesive Samples

The paper-adhesive samples were prepared in a four step process. First, the papers were trimmed to 210 mm × 99 mm to simplify the application of adhesive. All investigated adhesives were applied using the same application method (brushing), under the same conditions according to ISO 187:1990 [38] and then left to dry for 48 h. In the final step, paper sheets with the adhesive were cut into 100 mm × 15 mm stripes. Prepared paper-adhesive samples were used for SFE determination, SEM and FTIR analysis. Tested paper-adhesive samples are listed in Table 2. In addition, five paper stripes without adhesive were prepared for determination of SFE of paper.

2.3.2. Adhesive Joint Samples

According to ASTM D1876-08(2015)e1 standard [39], for adhesive joint samples preparation two paper sheets were trimmed to 210 mm × 70 mm and then bonded together only over 40 mm of their length to form T-peel test panel. After bonding, the end pressure (3 Pa) was applied for 1 h using pressing board with the weight and then dried for 48 h. After drying, the panels were cut into 25 mm wide test strips to form standard T-peel samples (Figure 2). For each tested sample groups fourteen adhesive joint T-peel samples were prepared, for machine grain direction (MD) and cross grain direction (CD) of paper. Adhesives were applied with the same application method (brushing) under the same standard conditions (conditioned for 7 days at a relative humidity of 50 ± 2% at 23 ± 1 °C). Tested adhesive joints are listed in Table 2.

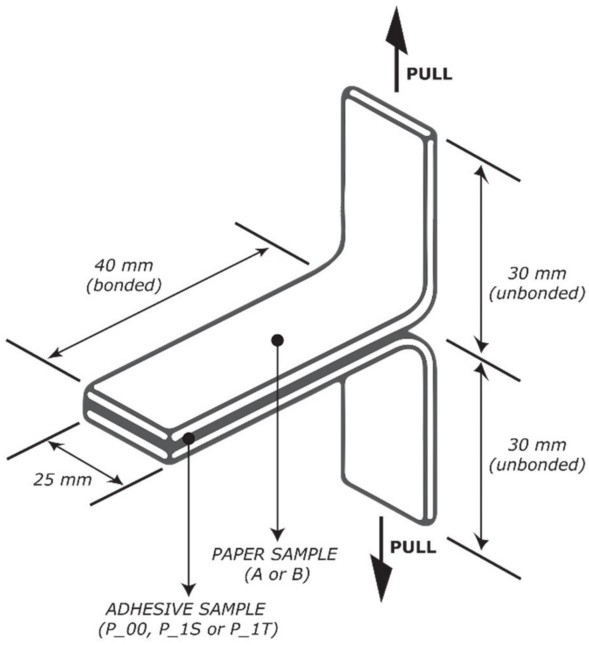

**Figure 2.** Schematic representation of T-peel test sample.

## 2.4. Determination of Surface Free Energy (SFE)

For determination of surface free energy (SFE) the Sessile Drop method on DataPhysics OCA 30 Goniometer (DataPhysics Instruments, Filderstadt, Germany) was used. By measuring the contact angles ($\theta$) between the solid surface and three different test liquids, with known surface tensions (demineralized water, diiodomethane and glycerol), free surface energies of papers and paper-adhesive samples were calculated using automatic Owens, Wendt, Rabel & Kaelble (OWRK) calculation method integrated in the SCA20 software (Version 2.01). OWRK method is the most frequently used method for determination of SFE ($\gamma$) of polymer surfaces [40], along with their polar ($\gamma^p$) and dispersive ($\gamma^d$) part. The droplet volume of tested liquids was 1 $\mu$L. Their contact angles were captured by CCD camera and measured after the initial contact of droplet with the sample (T 558 om-15) [41].

## 2.5. Scanning Electron Microscopy (SEM Microscopy)

Surfaces of the paper-adhesive samples were observed with SEM microscope Tescan Vega 3 (20 kV) (Tescan Orsay Holding, Brno, Czech Republic), under magnification of 2000×. Before examining, the samples were sputter coated with a thin layer of Pt/Pd.

## 2.6. Fourier Transform Infrared Spectroscopy (FTIR Spectroscopy)

The ATR spectra of the paper-adhesive samples were measured using Shimadzu FTIR IRAffinity-21 spectrometer (Shimadzu Corporation, Nishinokyo, Japan), with the Specac Silver Gate Evolution as a single reflection ATR sampling accessory with a ZnSe flat crystal plate (index of refraction 2.4). The IR spectra were recorded in the spectral range between 4500 and 500 cm$^{-1}$ at 4 cm$^{-1}$ resolution and averaged over 15 scans.

## 2.7. Peel Resistance of Adhesive (T-Peel Test)

T-peel resistance was conducted as an end use test, in addition to above described methods used to investigate the interactions between the substrate and the adhesive. T-peel test simulates the type of loading and service conditions to which a joint will be subjected. It is also used for comparison of peel resistance of different adhesives between the same materials. According to ASTM D1876-08(2015)e1 standard [39], peel resistance is described as the average force per unit width, measured along the bond line that is required to separate the bonded joint. T-peel resistance was measured using Mark 10 ES30 stand in combination with digital force gauge (Mark-10 Corporation, Copiague, NY, USA) and parallel jaw grips G1015–1 (Mark-10 Corporation, Copiague, NY, USA).

## 3. Results and Discussion

### 3.1. Properties of Paper Samples

The properties of analyzed paper samples, obtained according to standard methods, are listed in Table 3.

**Table 3.** Properties of paper samples.

| Paper Sample | Grammage (g/m$^2$) | Roughness (μm) | Moisture (%) | CaCO$_3$ (%) | Ash (%) | Cobb (g/m$^2$) | Tensile Strength (kN/m) |
|---|---|---|---|---|---|---|---|
| A | 80 | 2.56 ± 0.001 | 4.78 ± 0.52 | 20.55 ± 2.07 | 13.67 ± 0.79 | Side 1: 41.43 ± 1.63 <br> Side 2: 41.27 ± 2.52 | 6.08 ± 0.18 [MD] <br> 2.28 ± 0.09 [CD] |
| B | 80 | 3.04 ± 0.005 | 4.40 ± 0.19 | 30.01 ± 0.27 | 16.63 ± 0.11 | Side 1: 31.86 ± 1.86 <br> Side 2: 32.30 ± 1.39 | 3.93 ± 0.23 [MD] <br> 1.66 ± 0.09 [CD] |

[MD] Machine grain direction of paper, [CD] Cross grain direction of paper.

In addition, in Table 4, contact angle ($\theta$) measurements between solid paper surface and three different test liquids with known surface tensions are listed as well.

**Table 4.** Contact angle (θ) measurements with test liquids on paper samples.

| Paper Sample | Contact Angle (°) | | |
|:---:|:---:|:---:|:---:|
| | **Water** | **Diiodomethane** | **Glycerol** |
| A | 110.9 ± 0.2 | 44.0 ± 0.5 | 89.0 ± 0.4 |
| B | 112.2 ± 0.1 | 42.5 ± 0.6 | 94.3 ± 0.6 |

Properties of two tested papers are very similar in the terms of roughness and moisture content. Based on the paper properties from Table 3 and high wettability with non-polar liquid (diiodomethane) (Table 4), it can be concluded that both papers have hydrophobic surface, but paper A is more hydrophilic and has higher absorptivity according to the Cobb test. This behavior of paper A can be described due to a greater number of polar interactions between water and cellulose, i.e., immediately formed strong hydrogen bonds between water and accessible OH groups in cellulose [42–44]. This claim is related to a higher mechanical strength of paper A, in both grain directions, which points to a higher proportion of cellulose fibers in paper A [45–47].

Today, the most widely used filler in the paper industry is calcium carbonate ($CaCO_3$). Increased amount of $CaCO_3$ in paper can reduce the surface free energy of interphase ($\gamma_{12}$) in PVAc adhesive paper joints, due to the establishment of coordinate bonds between the acetate groups and $Ca^{2+}$ ions [43,48]. Higher amount of filler can reduce the stiffness and strength of the paper [42]. Higher content of $CaCO_3$ in paper B can also be related to lower absorptivity and its higher surface hydrophobicity [45,49,50]. Today's office papers are super calendared to avoid a strong difference in layer orientation between the top and bottom side of the sheet. Insufficient surface strength and inadequately fixed filler on the paper surface can result in dust accumulation and contamination of digital printing and copy machines [45].

### 3.2. Adhesion Performance Based on Surface Free Energies

After automatic calculation of SFE in polar and dispersive components of paper and paper-adhesive samples in the SCA20 software, dispersive ($x^d$) and polar ($x^p$) indexes were calculated as well (Table 5). When values of the polar and dispersive parts, i.e., polar and dispersive indexes of two phases are closer, more interactions are possible between these two phases and better adhesion is to be expected [51]. The dispersive component had the major contribution to the total surface free energy for all samples, while the polar component was significantly higher for the P_00/A paper-adhesive sample (39.099%). In addition, all paper A adhesive samples had a higher polar index then paper B adhesive samples. According to non-polar, hydrophobic paper surface characteristics, better adhesion performance was expected for paper B adhesive samples.

The optimal adhesion at the interface of paper and adhesive can be predicted by calculation of surface parameters of the paper-adhesive samples. The work of adhesion ($W_{12}$) is given by the Equation (1) [52]:

$$W_{12} = \gamma_1 + \gamma_2 - \gamma_{12} \tag{1}$$

where the subscript refers to surface free energy of each phase, and the $\gamma_{12}$ denotes their $\gamma$ of the interphase. The SFE of interphase ($\gamma_{12}$) was calculated using Equation (2) based on the two surface free energies ($\gamma_1$ and $\gamma_2$) and the similar interactions between the phases. These interactions are interpreted as the geometric mean of a dispersive part ($\gamma^d$) and a polar part ($\gamma^p$) of the SFE [9,53].

$$\gamma_{12} = \gamma_1 + \gamma_2 - 2\left( \sqrt{\gamma_1^d \times \gamma_2^d} + \sqrt{\gamma_1^p \times \gamma_2^p} \right) \tag{2}$$

The wetting coefficient ($S_{12}$), which is measure of the tendency of a liquid phase to spread on another liquid or solid phase, was calculated using Equation (3). If the coefficient is positive, the liquid phase will spread. If the coefficient is negative, wetting will not be complete [9].

$$S_{12} = \gamma_1 - \gamma_2 - \gamma_{12} \tag{3}$$

In addition to the work of adhesion, the SFE of interphase and wetting coefficient and the difference between paper and paper-adhesive sample dispersive index was calculated ($x_2{}^d - x_1{}^d$) as well (Table 6).

**Table 5.** Surface free energies ($\gamma$), polar ($\gamma^p$) and dispersive ($\gamma^d$) components, dispersive and polar indexes ($x^d$, $x^p$) of papers and paper-adhesive samples according to OWRK method.

|  |  | $\gamma$ (mJ/m$^2$) | $\gamma^p$ (mJ/m$^2$) | $\gamma^d$ (mJ/m$^2$) | $x^d$ (%) | $x^p$ (%) |
|---|---|---|---|---|---|---|
| Paper Sample | A | 37.32 | 0.55 | 36.77 | 98.526 | 1.474 |
|  | B | 36.83 | 0.81 | 36.02 | 97.800 | 2.199 |
| Paper-adhesive sample | P_00/A | 51.97 | 20.32 | 31.65 | 60.900 | 39.099 |
|  | P_1S/A | 45.02 | 10.83 | 34.19 | 75.944 | 24.056 |
|  | P_2S/A | 46.38 | 11.11 | 35.28 | 76.067 | 23.954 |
|  | P_3S/A | 49.65 | 16.54 | 33.11 | 66.687 | 33.313 |
|  | P_1T/A | 48.39 | 11.44 | 36.94 | 76.338 | 23.641 |
|  | P_2T/A | 47.14 | 13.41 | 33.73 | 71.553 | 28.447 |
|  | P_3T/A | 46.22 | 12.60 | 33.64 | 72.739 | 27.260 |
|  | P_00/B | 39.47 | 2.98 | 36.49 | 92.450 | 7.550 |
|  | P_1S/B | 41.01 | 2.39 | 38.61 | 94.148 | 5.828 |
|  | P_2S/B | 41.54 | 4.05 | 37.49 | 90.250 | 9.750 |
|  | P_3S/B | 42.95 | 4.78 | 38.16 | 88.847 | 11.130 |
|  | P_1T/B | 42.78 | 5.47 | 37.31 | 87.214 | 12.786 |
|  | P_2T/B | 47.21 | 12.8 | 34.41 | 72.887 | 27.113 |
|  | P_3T/B | 45.37 | 8.20 | 37.16 | 81.904 | 18.073 |

**Table 6.** Adhesion parameters for paper-adhesive samples: work of adhesion ($W_{12}$), SFE of interphase ($\gamma_{12}$), wetting coefficient ($S_{12}$) and difference between paper and paper-adhesive dispersive index ($x_2{}^d - x_1{}^d$).

|  |  | $W_{12}$ (mJ/m$^2$) | $\gamma_{12}$ (mJ/m$^2$) | $S_{12}$ (mJ/m$^2$) | $x_2{}^d - x_1{}^d$ (%) |
|---|---|---|---|---|---|
| Paper-adhesive sample | P_00/A | 79.914 | 14.376 | −29.026 | 37.626 |
|  | P_1S/A | 75.794 | 6.546 | −14.246 | 22.582 |
|  | P_2S/A | 76.978 | 6.722 | −15.782 | 22.459 |
|  | P_3S/A | 75.816 | 11.154 | −23.484 | 31.839 |
|  | P_1T/A | 78.727 | 6.983 | −18.053 | 22.188 |
|  | P_2T/A | 75.866 | 8.594 | −18.414 | 26.973 |
|  | P_3T/A | 75.584 | 7.956 | −16.856 | 25.787 |
|  | P_00/B | 76.616 | 0.684 | −3.324 | 5.351 |
|  | P_1S/B | 77.368 | 0.472 | −4.652 | 3.653 |
|  | P_2S/B | 77.118 | 1.252 | −5.962 | 7.550 |
|  | P_3S/B | 78.085 | 1.695 | −7.815 | 8.953 |
|  | P_1T/B | 77.528 | 2.082 | −8.032 | 10.587 |
|  | P_2T/B | 76.851 | 7.189 | −17.569 | 24.914 |
|  | P_3T/B | 78.326 | 3.874 | −12.414 | 15.896 |

In order to obtain the optimal adhesion, SFE of interphase should be minimal (tends to zero), work of adhesion should be maximal and wetting coefficient should be as close to zero or positive [9,20,26,54–56].

Figure 3 shows an adhesion performance results and rank of all investigated combinations for the above mentioned parameters. Rankings of the three parameters ($\gamma_{12}$, $S_{12}$, $x_2{}^d - x_1{}^d$) were almost the same (in nine out of fourteen samples). It is important to emphasize that samples that do not match in all of these three parameters, do not show any major ranking deviations (max ± 3 places). For the fourth parameter ($W_{12}$) very small differences between obtained values were recorded.

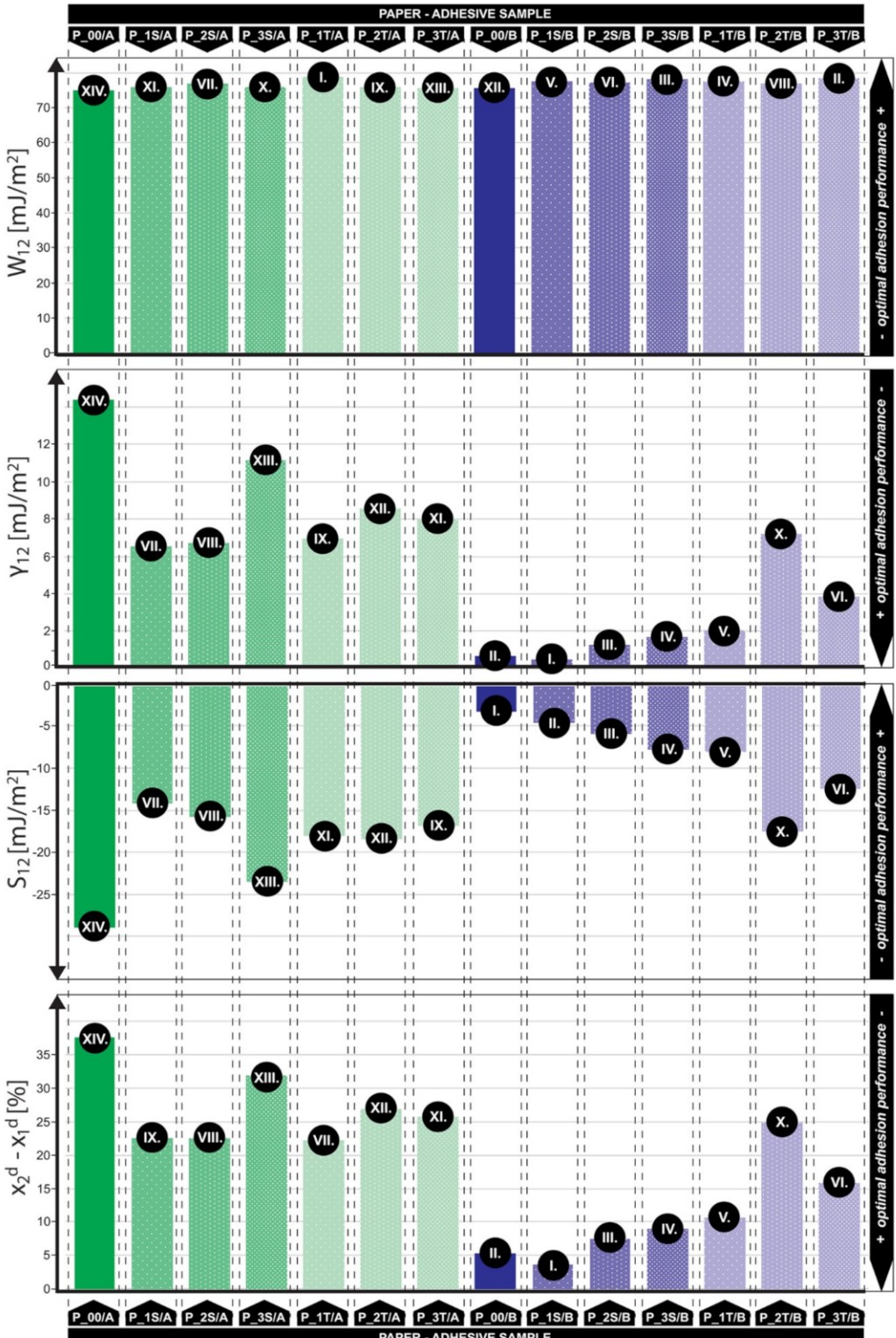

**Figure 3.** Adhesion performance results and rank according to work of adhesion ($W_{12}$), SFE of interphase ($\gamma_{12}$), wetting coefficient ($S_{12}$) and difference between paper and paper-adhesive dispersive index ($x_2{}^d - x_1{}^d$).

The presence of $SiO_2$ or $TiO_2$ nanoparticles in adhesive had affected the surface properties of all paper-adhesive samples. While the work of adhesion remained almost the same, better adhesion for all paper A adhesive samples and P_1S/B was achieved by lowering SFE of interphase and increasing

the wetting coefficient. Addition of nanoparticles affected the surface properties of paper-adhesive samples by lowering their polarity for all paper A adhesive samples and P_1S/B sample. According to the ranking from Figure 3, it can be assumed that optimum nanoparticles concentration of $SiO_2$ and $TiO_2$ nanoparticles in PVAc adhesive is 1% for both types of paper. At higher concentrations, the polarity was increased again, and therefore the difference between paper and paper-adhesive dispersive indexes as well.

The best adhesion performance was achieved with the addition of 1% of $SiO_2$ nanoparticles in PVAc adhesive. With the addition of 1% of $TiO_2$ nanoparticles, better adhesion performance compared to PVAc adhesive, was achieved only for paper A. The best adhesion performance can be predicted for P_1S/B sample.

### 3.3. SEM Microscopy

By comparing SEM micrographs of paper-adhesive samples for A and B paper with PVAc adhesive (Figure 4), it is clear that the surface of P_00/B sample was much smoother (Figure 4b). The surface of P_00/A sample showed many cavities in dry adhesive film (Figure 4a). It can be explained by a lack of interactions between PVAc adhesive and paper A, due to the much higher polarity of P_00/A sample, i.e., the high difference between paper and paper-adhesive dispersive index.

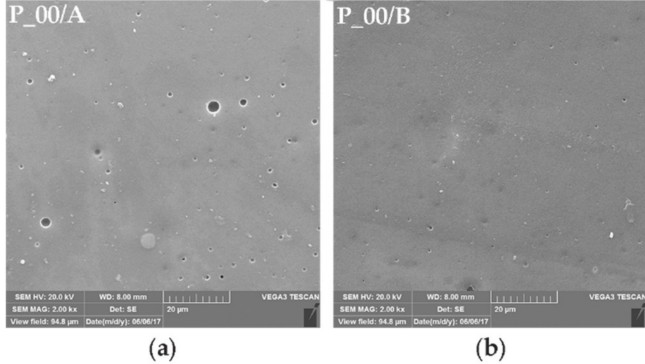

**Figure 4.** SEM micrographs of paper-adhesive samples for A and B paper with PVAc adhesive (magnification 2000×). (**a**) P_00/A; (**b**) P_00/B.

Nanoparticles have high tendency for particle aggregation and it is of particular importance to have a good distribution in the polymer matrix in order to achieve improvement of properties. The aggregated nanoparticles appear as white dots on SEM micrographs [26]. In addition, white dots, unsmooth or uneven surface may be associated with the high proportion of nanoparticles, but also with inadequate application or stirring technique. Figure 5 shows good distribution of nanoparticles in PVAc adhesive. Low particle aggregation is present on paper-adhesive surfaces with a higher concentration of nanoparticles (2% and 3%). Therefore, the previous claim, about optimum nanoparticles concentration based on SFE from Figure 3, is confirmed.

For investigated paper-adhesive combinations modified with nanoparticles, the best adhesion performance was achieved for samples with 1% of nanoparticles. Higher concentrations of nanoparticles led to a larger clustering and unwanted changes of desirable properties—increase of SFE of interphase, higher difference between paper and paper-adhesive dispersive index and shifting of wetting coefficient away from zero. By comparing the SEM micrographs of paper A PVAc adhesive (Figure 4a), and any other SEM micrographs of paper A with nano-enhanced PVAc adhesive (Figure 5a–f), it is obvious that surfaces with nano-enhanced adhesives are much smoother, more compact, and in addition there is no observed microphase separation.

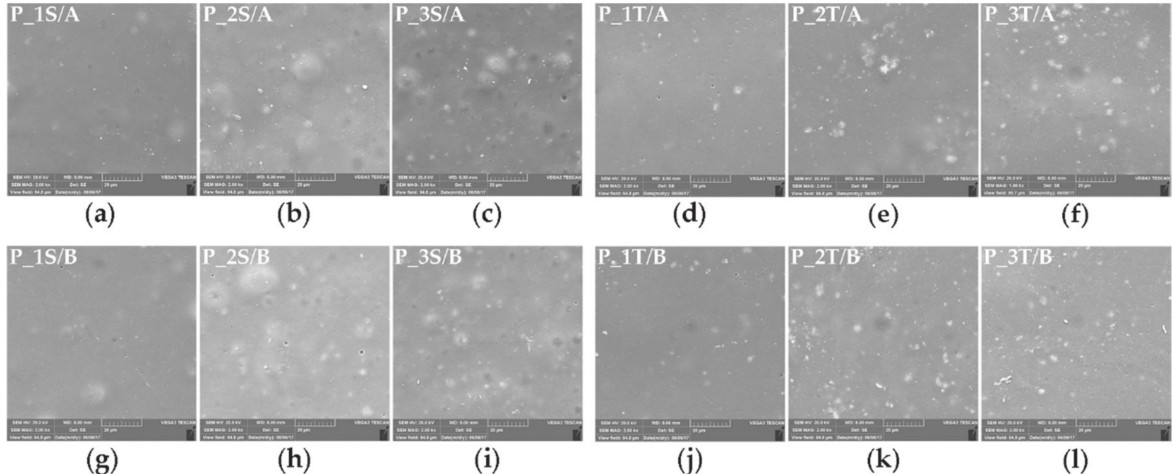

**Figure 5.** SEM micrographs of paper-adhesive samples with nano-enhanced PVAc adhesive (magnification 2000×). (**a**) P_1S/A; (**b**) P_2S/A; (**c**) P_3S/A; (**d**) P_1T/A; (**e**) P_2T/A; (**f**) P_3T/A; (**g**) P_1S/B; (**h**) P_2S/B; (**i**) P_3S/B; (**j**) P_1T/B; (**k**) P_2T/B; (**l**) P_3T/B.

### 3.4. FTIR Spectroscopy

Spectra of papers with PVAc nanocomposites that are recorded on the side of PVAc film on the papers are presented in Figures 6 and 7.

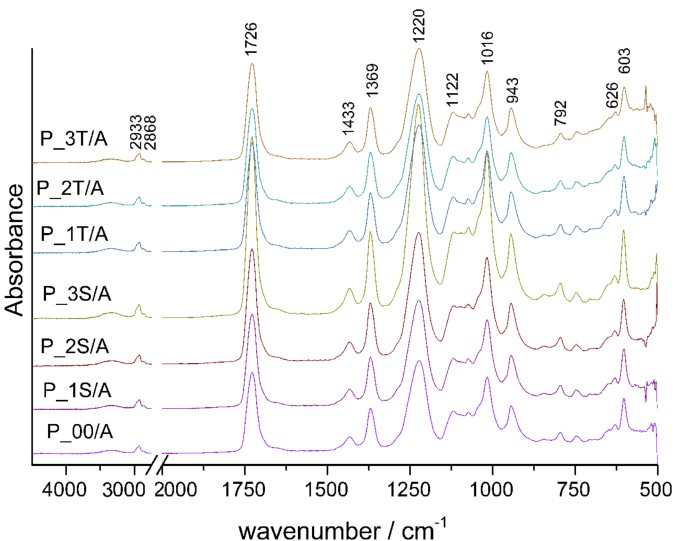

**Figure 6.** The FTIR spectra of paper A adhesive sample (P_00/A, P_1S/A, P_2S/A, P_3S/A, P_1T/A, P_2T/A, P_3T/A).

Bands ranging from 2933 to 2868 cm$^{-1}$ correspond to the CH, CH$_2$ and CH$_3$ group stretching vibrations of PVAc [57]. The ester carbonyl (C=O) stretching vibration associated to acetate groups with a molecular vibration at 1726 cm$^{-1}$ is complemented by two less intense peaks at 1433 and 1369 cm$^{-1}$ due to the CH$_3$ asymmetric and symmetric bending vibration in the case of paper A, respectively. In the case of paper B, the ester carbonyl vibrational band is shifted to 1730 cm$^{-1}$. The peak at 1220 cm$^{-1}$ corresponds to the asymmetric stretching mode of C–C(=O)–O ester group of PVAc, and it is followed by vibrational bands with the maximum at 1122 cm$^{-1}$, 1016 cm$^{-1}$ and the less intense one at 943 cm$^{-1}$, in the case of paper A. In the case of paper B, the bands are shifter towards 1224, 1120, 1018 and 945 cm$^{-1}$. Additionally, less intense peaks such as the C–H rocking vibration at 792, 626 and 603 cm$^{-1}$ in the case of paper A, and 794, 628 and 603 cm$^{-1}$ in the case of paper B were detected [58,59]. The FTIR spectra of PVAc paper-adhesive samples modified with SiO$_2$ and TiO$_2$ nanoparticles exhibit almost

the same characteristics vibrations as PVAc adhesive. The characteristic peaks of $SiO_2$ and $TiO_2$ nanoparticles were not identified. Considering the SEM micrographs and obtained FTIR spectra, it can be concluded that good uniform dispersion and existence of the good interactions of nanoparticles and PVAc adhesive was achieved. Additionally, due to coverage of nanoparticles with the adhesive, the vibrational bands of nanoparticles are probably covered and overlapped with vibrational bands of adhesive which is present in a larger amount. Considering that FTIR spectra did not show any significant differences in PVAc nanocomposites with regard to the share of nanoparticles, the further T-peel resistance was done only for the PVAc nanocomposites with 1% of the nanoparticles.

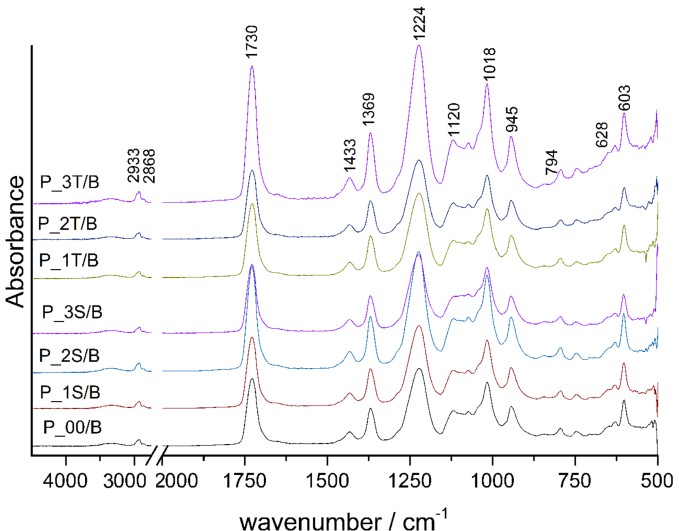

**Figure 7.** The FTIR spectra of paper B adhesive sample (P_00/B, P_1S/B, P_2S/B, P_3S/B, P_1T/B, P_2T/B, P_3T/B).

### 3.5. End Test of Adhesive Joints Strength According to T-peel Resistance

The peel resistance depends on many factors such as the peeling angle, the nature of adhesive, mechanical and physical properties of the substrate, temperature and humidity of the environment, the conditioning process and the cohesive properties of the interface [60,61]. The peel test quantifies the strength of the adhesive joint, but normally it is used to compare adhesives. The peel test provides useful comparative data, but not quantitative measure of interface strength [62,63]. The average peel resistance values obtained are given in Table 7 as mean ± SD. Twelve different adhesive joint combinations were tested and results are compared in Figure 8.

Figure 8 shows that the highest peel resistance of adhesive joints for both papers was achieved with PVAc adhesive modified with 1% of $SiO_2$ nanoparticles. According to obtained results in Table 7, resistance improvement for adhesive joints with P_1S adhesive was between 13.46%–23.88% (A/P_1S/A:15.38%(MD), 23.88%(CD); B/P_1S/B:17.10%(MD), 13.46%(CD)), respectively. Modification of PVAc with 1% of $TiO_2$ nanoparticles did not significantly influence the peel resistance. Adhesive joints formed with P_1T adhesive and paper A had a very small increase in peel resistance (1.71%(MD), 4.48 %(CD)), while joints with P_1T and paper B had a small decrease of peel resistance. The highest peel resistance was obtained for A/P_1S/A(MD) sample (270 N/m), while the lowest peel resistance was obtained for B/P_1T/B(CD) sample (94 N/m).

Considering that one of the main factors for peel resistance are mechanical and physical properties of the substrate, it was expected that the samples with substrate A would have higher peel resistance because of higher tensile strength (Table 7). Adhesion performance test results based on SFE, with the ranking shown in Figure 3, correspond to T-peel test results within a same group of paper and both grain directions.

**Table 7.** Adhesive joints peel resistance with paper tensile strength, for MD and CD paper grain direction, and improvement of peel resistance for joints with nano-enhanced adhesives.

| Grain Direction | Paper Sample | Tensile Strength (kN/m) | Adhesive Joint Sample | Peel Resistance (N/m) | Peel Improvement (%) |
|---|---|---|---|---|---|
| MD | A | 6.08 ± 0.18 | A/P_00/A | 234.00 ± 42.00 | Ref. |
| | | | A/P_1S/A | 270.00 ± 44.05 | 15.38 |
| | | | A/P_1T/A | 238.00 ± 36.28 | 1.71 |
| | B | 3.93 ± 0.23 | B/P_00/B | 152.00 ± 18.33 | Ref. |
| | | | B/P_1S/B | 178.00 ± 26.00 | 17.10 |
| | | | B/P_1T/B | 150.00 ± 22.36 | −1.31 |
| CD | A | 2.28 ± 0.09 | A/P_00/A | 134.00 ± 20.10 | Ref. |
| | | | A/P_1S/A | 166.00 ± 28.36 | 23.88 |
| | | | A/P_1T/A | 140.00 ± 12.65 | 4.48 |
| | B | 1.66 ± 0.09 | B/P_00/B | 104.00 ± 14.97 | Ref. |
| | | | B/P_1S/B | 118.00 ± 18.87 | 13.46 |
| | | | B/P_1T/B | 94.00 ± 9.17 | −9.61 |

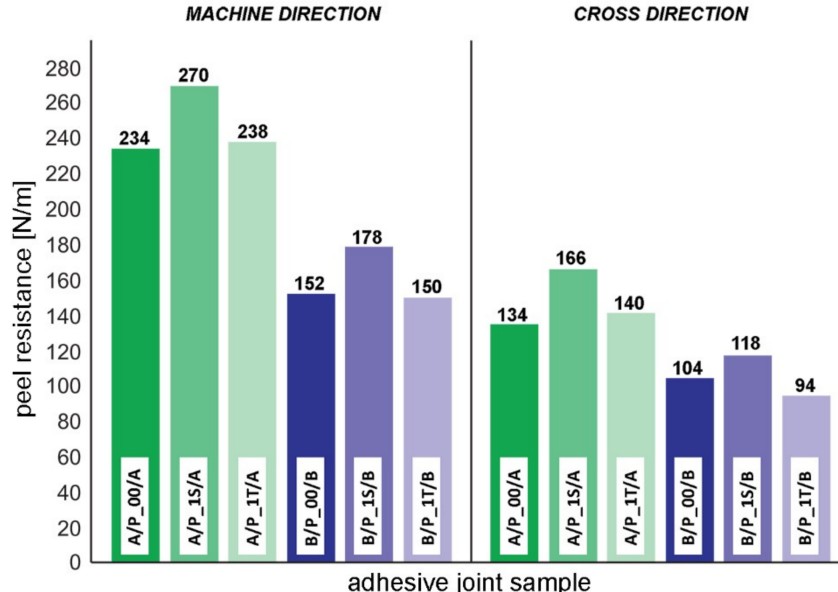

**Figure 8.** Comparison of the obtained results for peel resistance of adhesive joint samples—machine grain direction (MD) and cross grain direction (CD).

## 4. Conclusions

Polymer nanocomposites of PVAc adhesive with 1%, 2% and 3% of $SiO_2$ and $TiO_2$ nanoparticles were manufactured to investigate the adhesion performance and strength of paper-adhesive joints. Uniform nanoparticles dispersion and effective mixing approach was confirmed with SEM microscopy and FTIR spectroscopy characterization. According to SEM micrographs, paper surfaces with nano-enhanced adhesive were smooth, compact and without observed microphase separation. Low particle aggregation was visible only on paper surfaces with nano-enhanced adhesive with a higher concentration of nanoparticles. Almost the same characteristic vibrations and peaks were identified for the FTIR spectra of paper with PVAc and for paper with nano-enhanced adhesives, without any significant differences in regard to the concentration of nanoparticles. In addition, the adhesion performance based on the surface free energies confirmed that 1% is optimum nanoparticles concentration for these particular paper-adhesive combinations. Addition of nanoparticles affected the surface properties of paper-adhesive samples by lowering their polarity for all paper A adhesive samples and paper B samples with PVAc adhesive modified with 1% of $SiO_2$, i.e., lowering the SFE of

interphase and increasing the wetting coefficient. The wettability of the paper-adhesive surface and low SFE of interphase turned out as a key for a good adhesion performance, since the work of adhesion remained almost the same for all tested samples. The end use strength test (T-peel) of adhesive joints indicated that nanoparticles can considerably improve the bonding strength in graphic post press production processes. The highest strength improvement was achieved with the addition of 1% of $SiO_2$ nanoparticles in PVAc adhesive (13.46%–23.88%). Modifications with 1% of $TiO_2$ nanoparticles did not have significant influence on the bonding strength. More specifically, adhesive joints with paper B and PVAc adhesive modified with 1% of $TiO_2$ even had a small decrease of peel resistance (−1.31%(MD), −9.61%(CD)). The highest peel resistance was obtained for A/P_1S/A(MD) sample (270 N/m), while the lowest peel resistance was obtained for B/P_1T/B(CD) sample (94 N/m). Adhesion performance test results based on SFE can be corresponded to T-peel test results within a same group of paper.

**Author Contributions:** Conceptualization, data curation, formal analysis, methodology, validation, writing – original draft, G.P. and M.V.; funding acquisition, project administration, G.P., M.V. and S.P.P.; investigation, resources, G.P., M.V. and J.B.; supervision, M.V. and S.P.P.; visualization, G.P.; writing – review & editing, J.B. and S.P.P.

**Funding:** This research was funded by the University of Zagreb, Grant under the title: "Modification of conventional graphic materials with nanoparticles and chromogenic materials and their health safety."

**Conflicts of Interest:** The authors declare no conflicts of interest.

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
