# Peer review of "Enhancement of Polyvinyl Acetate (PVAc) Adhesion Performance by SiO2 and TiO2 Nanoparticles"

_coatings, doi:10.3390/coatings9110707_

Round 1

Reviewer 1 Report

Line 82 – Table was not mentioned yet, I would recommend using the same numbering of Tables as they are mentioned in the text.

Line 87 – I am missing the information about material properties of selected adhesives.

Line 93 – In my opinion, the name of the section is confusing. The adhesives had been already prepared; you are modifying the existing adhesive.

Author Response

Dear Reviewer,

Please find in the attachment "Responses to Reviewer Comments".

Best regards,

The Authors

Comment: Line 82 – Table was not mentioned yet, I would recommend using the same numbering of Tables as they are mentioned in the text.

Response: Table is related to the results arising from the used methods mentioned in the paragraph (lines 117-122). Thus, the sentence “The results of all paper characterizations are listed in Table 2.” was deleted. Table with properties of paper samples (now Table 3) is still presented and described in the Results and Discussion section (lines 199-202).

Comment: Line 82 – I am missing the information about material properties of selected adhesives.

Response: Additional table was added in the manuscript as “Table 1. Properties of Signokol L adhesive given by the producer“ (lines 129-130). In order to avoid repetition of the information, part of the sentence in lines 127-128 (45 ± 2% of solid content and transparent dry film color) was deleted. 

Comment: Line 82 – In my opinion, the name of the section is confusing. The adhesives had been already prepared; you are modifying the existing adhesive.

Response: The correction was made according to reviewer suggestion. Now the section is named as “Modification of PVAc Adhesive” (line 136).

Reviewer 2 Report

This paper describes a series of systematic and comparative experiments on the effect of SiO2 and TiO2 nanoparticles on structural and adhesion properties of PVAc-based coatings. This manuscript is in well-written. The experimental results are interesting with a finding of the optimal filler content of 1 wt%, which has certain engineering significance. However, from my point of view, the main shortcoming of this manuscript is the lack of scientific discussion with deeper insights into the enhancement mechanisms. Prior to publication as a RESEARCH article in Coatings, the authors need to present their own understandings and opinions rather than just conclude the experimental results. The explanations and understandings might be more beneficial for the readers. Several issues I have are listed below.

Too many keywords including “nano”. Introduction part; Paragraph 3: Since the researchers have been worked on the silica and titanium dioxide nanoparticles, what is the difference and novelty of this work when compared with the previous findings? Introduction part; Paragraph 4: Fourier Transform Infrared Spectroscopy (FTIR) should be Fourier Transform Infrared (FTIR) Spectroscopy. Again, the authors are suggested to state the reasons for selecting these two nanoparticles at the beginning of this manuscript as well as to compare and discuss & conclude the possible difference influence mechanisms such as surface states? crystal structures? et al. between these two fillers. Section 3.4: Paragraph 2: The sentence “The characteristic peaks of SiO2 and TiO2 nanoparticles were not identified, which indicates good uniform dispersion and existence of the good interactions of nanoparticles and PVAc adhesive” confused me a bit. First, is there no characteristic peaks of Si-O-Si and Ti-O-Ti? Additionally, how did the authors conclude that the characteristic peaks are associated with the dispersion of fillers?

To conclude, this work is interesting and will certainly benefit the audiences of Coatings. I hope the authors could present their own understandings and opinions in the revised version.

Author Response

Dear Reviewer,

Please find in the attachment "Responses to Reviewer Comments".

Best regards,

The Authors

Comment: Too many keywords including “nano”.

Response: The corrections were made according to reviewer suggestion. Two keywords including “nano” prefix were deleted. Now, the Keywords are listed as: PVAc adhesive; nano-enhanced adhesive; adhesive joint; adhesion performance; peel resistance (lines 24-25).

Comment: Introduction part; Paragraph 3: Since the researchers have been worked on the silica and titanium dioxide nanoparticles, what is the difference and novelty of this work when compared with the previous findings?

Response: The corrections were made according to reviewer suggestion and included in the manuscript (lines 67-73): Based on previous research, it can be concluded that modification of PVAc adhesive with NC, CNFs, SiO2 or TiO2 nanoparticles has not been explored enough for the purpose of postpress processes in graphic production. The existing studies are mainly related with the improvement of bonding performance in wood and furniture industry [16,18,21,23,24,27]. Considering that the performance of adhesive depends upon the nature of adherends [26], the existing results cannot be directly linked to the adhesive systems within the graphic production, more precisely, within the production of high quality paper products.

Comment: Introduction part; Paragraph 4: Fourier Transform Infrared Spectroscopy (FTIR) should be Fourier Transform Infrared (FTIR) Spectroscopy.

Response: The correction was made according to reviewer suggestion in line 77.

Comment: Again, the authors are suggested to state the reasons for selecting these two nanoparticles at the beginning of this manuscript as well as to compare and discuss & conclude the possible difference influence mechanisms such as surface states? crystal structures? et al. between these two fillers.

Response: The corrections were made according to reviewer suggestion and included in the manuscript (lines 78-113): “In this research, SiO2 and TiO2 nanoparticles were selected according to recommendations and results of previous studies on polymer matrix nanocomposites (PMCs) [16-19,21-25,27]. According to their remarkable properties and acceptable costs, SiO2 nanoparticles are recommended for the production of high-performance adhesives and coatings. In addition, they can be used for the enhancement of mechanical strength, flexibility, durability, as well as modification of rheological properties of liquids, adhesives and elastomers [17,25,27]. TiO2 nanoparticles have been rarely used in PMCs modification studies, due to weaker mechanical strength results compared to SiO2 nanoparticles. However, the addition of TiO2 nanoparticles can reduce material degradation under the influence of UV radiation, ensure consistence of coloration and to increase the lifetime of the final products [25]. The invisibility or whiteness of the adhesive bond line can be very important for the appearance of the final graphic product. TiO2 pigmentary properties impart whiteness, brightness and opacity when incorporated into PMCs. By increasing the lifetime of the final product, adhesive bonding processes can become even more important in graphic postpress processes for the production of more competitive and more durable products compared to other joining methods (e.g. sewing, stapling, riveting). NC is still the most studied nanomaterial due to low cost production and availability, but mostly it is used in the construction industry for modification of concrete, cement, asphalt and bitumen to enhance their barrier, mechanical and rheological properties, fire retardancy, as well as their liquid infusion resistant properties [24,25]. Addition of NC or CNFs can cause an increase in viscosity [19,25]. Moreover, addition of NC can cause yellowing [24]. The mentioned changes are not desirable when using PVAc adhesive in graphic postpress processes, regardless to the application methods and appearance of the final product. It is expected that the advantages obtained by addition of relative expensive nanoparticles will compensate the increase of the adhesive price [26]. Therefore, it is of great interest to keep the share of nanoparticles in the selected adhesive as small as possible. According to previous studies, it is possible to achieve the best performance of selected PMCs with a smaller share of SiO2 or TiO2 nanoparticles (1%) [16] compared to NC (4%) [18] and CNFs (10%) [19]. Considering the fact that the results of conducted studies are not related to PVAc adhesive systems and mutually differ, higher increase in mechanical strength of adhesives can be achieved with SiO2 nanoparticles then NC or CNFs. For example, the epoxy adhesive mechanical strength was increased up to 66% with SiO2 nanoparticles, with CNFs up to 5% and with NC up to 7%, respectively [26]. The mechanical strength of polyurethane adhesive was increased up to 462% with SiO2 nanoparticles compared to 68% with NC, while the increase of acrylic adhesive was up to 219% with SiO2 nanoparticles compared to 146% with NC [26]. By addition of SiO2 and TiO2 nanoparticles, besides mechanical properties, thermal stability and durability improvement, considerable increase of the bonding strength of PVAc adhesive, at open time 5 and 10 minutes, can be achieved as well [21], which can lead to higher efficiency of bonding processes in graphic production.”

Comment: Section 3.4: Paragraph 2: The sentence “The characteristic peaks of SiO2 and TiO2 nanoparticles were not identified, which indicates good uniform dispersion and existence of the good interactions of nanoparticles and PVAc adhesive” confused me a bit. First, is there no characteristic peaks of Si-O-Si and Ti-O-Ti? Additionally, how did the authors conclude that the characteristic peaks are associated with the dispersion of fillers?

Response: Characteristic peaks of SiO2 and TiO2 were not identified. Based on the obtained SEM micrographs and FTIR spectra we assumed that good dispersion of nanoparticles was achieved. Additionally, due to relatively small share of nanoparticles in PVAc adhesive, it may be assumed  that vibrational bands associated with nanoparticles were overlapped or covered by strong bands of the adhesive, not allowing assignment of the bands solely to the nanoparticles. Thus, additional sentences were added in the manuscript (lines 327-332) as follows: The characteristic peaks of SiO2 and TiO2 nanoparticles were not identified. Considering the SEM micrographs and obtained FTIR spectra, it can be concluded that good uniform dispersion and existence of the good interactions of nanoparticles and PVAc adhesive was achieved. Additionally, due to coverage of nanoparticles with the adhesive, the vibrational bands of nanoparticles are probably covered and overlapped with vibrational bands of adhesive which is present in a larger amount.“

Comment: To conclude, this work is interesting and will certainly benefit the audiences of Coatings. I hope the authors could present their own understandings and opinions in the revised version.

Response: We hope that the revised version of the manuscript will meet the Journal’s publication requirements.

Round 2

Reviewer 2 Report

The authors have satisfactorily replied to all my queries, therefore, this paper can be considered for publication in Coatings.